# How to Data in Datathons

**Carlos Mougan**
The Alan Turing Institute
University of Southampton

**Richard Plant**
The Alan Turing Institute
Edinburgh Napier University

**Clare Teng**
The Alan Turing Institute

**Marya Bazzi**
The Alan Turing Institute
University of Warwick

**Alvaro Cabrejas Egea**
Fujitsu Research of Europe

**Ryan Sze-Yin Chan**
The Alan Turing Institute

**David Salvador Jasin**
The Alan Turing Institute

**Martin Stoffel**
The Alan Turing Institute

**Kirstie Jane Whitaker**
The Alan Turing Institute

**Jules Manser**
The Alan Turing Institute
`jmanser@turing.ac.uk`

## Abstract

The rise of datathons, also known as data or data science hackathons, has provided a platform to collaborate, learn, and innovate in a short timeframe. Despite their significant potential benefits, organizations often struggle to effectively work with data due to a lack of clear guidelines and best practices for potential issues that might arise. Drawing on our own experiences and insights from organizing $\geq 80$ datathon challenges with $\geq 60$ partnership organizations since 2016, we provide guidelines and recommendations that serve as a resource for organizers to navigate the data-related complexities of datathons. We apply our proposed framework to 10 case studies.

## 1 Introduction

Datathons or data hackathons, loosely defined as data or data science-centric hackathons (Anslow et al., 2016; Chau and Gerber, 2023), have become increasingly popular in recent years, providing a platform for participants and organisations to collaborate, innovate, and learn in the area of data science over a short timeframe. This paper focuses on data-related challenges in datathons, including: What does it mean for data to be "appropriate" for a datathon? How much data is "enough" data? How can we identify, categorise, and use "sensitive" data? We aim to offer a set of guidelines and recommendations to prepare different types of data for datathons drawn from extensive experience in datathon organisation.

Working with and preparing data for a datathon can be challenging due to problem-specificity and the short time scales available to prepare the data for analysis and answer a specific set of questions. In research projects, there is significantly more time and room to explore and adapt the data and the questions. Other factors include the type of datathon, its potential breadth (e.g., data which is publicly available and has been previously analysed by researchers (Kitsios and Kamariotou, 2019; Concilio et al., 2017)), datasets from a company with no or limited prior analysis (Nolte et al., 2018; Nolte, 2019), or working with multiple data sources and formats. To effectively incorporate (varied) data, there is a need to have standardised processes and definitions that can account for the complexity and challenges that arise during the datathon selection and preparation process. In particular, our

37th Conference on Neural Information Processing Systems (NeurIPS 2023) Track on Datasets and Benchmarks.

paper differentiates and focuses on the following data-related dimensions: appropriateness, readiness, reliability, sensitivity and sufficiency. Our main contributions are as follows:

- Introduce a framework to analyse the data intended for datathons from several dimensions: appropriateness, readiness, reliability, sensitivity, and sufficiency.
- Share insights and experiences from organising datathons with external organisations, mapping 10 data challenges and organisations to the proposed framework.
- Provide recommendations and best practices for organisers to effectively select, prepare, and categorise data for datathon challenges.

Our paper is organised as follows. In section 2, we overview various aspects of hackathons and datathons, data assessment terminology and frameworks. In section 3, we describe Data Study Groups (DSGs), which form the methodology for this paper. In section 4, we introduce our data assessment framework, which we map to existing terminology in the literature, where appropriate. Section 5 details ten datathon use cases and shows how these can be mapped to the proposed framework. In particular, we map the three most recently completed DSGs to the framework we propose. We offer some general recommendations on improving data quality in datathons as an event organiser in section 6. Finally, section 7 provides a summary of the contribution.

## 2 Related Work

### 2.1 Hackathons & Datathons

Previous research on hackathons has provided valuable insights that can enhance one's understanding of datathons and their effects on organisations and participants. Extensive literature reviews include Chau and Gerber (2023); Olesen and Halskov (2020). Chau and Gerber (2023) found that hackathon and datathon research tends to focus on four core areas: the event purpose, the event format, the processes undertaken and experienced by stakeholders, and the event outcomes (with some papers having more than one focus). They also add that most publications about processes focus on what happens during the event (e.g., intra-team dynamics in collaboration and project brainstorming) rather than before or after.

As mentioned in the introduction, datathons are data science or data-centric hackathons. The work in Anslow et al. (2016) explores how these can enhance data science curriculum (e.g., through improved data science, communication, and community engagement skills) and that in Kuter and Wedrychowicz (2020) on how to run a datathon with very limited resources. Aboab et al. (2016) focus on healthcare and emphasize the cross-disciplinary aspects of datathons, claiming they help address poor study design, paucity of data, and improve analytical rigour through increased transparency. Luo et al. (2021) conducted a virtual datathon focusing on the early stages of the Covid-19 pandemic, highlighting the importance of interdisciplinary and diverse team compositions. Piza et al. (2018) found a significant association between positive teamworking behaviors and affective learning in a healthcare-focused datathon, pointing out effective leadership as the key factor.

We propose to add to existing literature that explores challenges and best practices in datathons, with a focus on data practices. More specifically, and following the terminology in Chau and Gerber (2023), we focus on data as a "technical project material" in datathon formats, and examine key challenges that can arise in data evaluation and preparation during the "pre-hackathon" organisation process. To the best of our knowledge, prior research has not focused on data-related challenges in the specific context of datathons.

### 2.2 Data Analysis Dimensions

Previous research on data quality evaluation has focused on general description frameworks and metrics, rather than datathon context-specific development of methods (Wang and Strong, 1996; Zaveri et al., 2016; Juran et al., 1974; Schlegel et al., 2020).

Early work in the area (Wang and Strong, 1996) presented a conceptual framework for assessing "data quality" (i.e. assessing issues that can affect the potentiality of the applications that use the data), which included 15 dimensions grouped under four categories: intrinsic; contextual; representational and system-supported. Indeed, "data quality" is commonly viewed as a multi-dimensional construct

that defines "fitness of use" and may depend on various factors (Zaveri et al., 2016; Juran et al., 1974), such as believability, accuracy, completeness, relevancy, objectivity, accessibility, amongst several others (Wang and Strong, 1996). The context-specific methodology of Schlegel et al. (2020) focuses on defect diagnosis and prediction and addresses the following five dimensions, which they argued were the most important for assessing "data quality" (which they rather refer to as "data suitability"): relevancy; completeness; appropriate amount (of data); accessibility and interpretability. Their study focused on developing guidance for companies, using various metrics and evaluation levels, on how to gain insight into the suitability of the data collected in the context of defect diagnosis and prediction. Our work extends and builds upon these previous frameworks and methodologies by applying them specifically to the challenges and requirements of datathons. In particular, we consider the five following dimensions in the context of data-driven problem-solving during the datathon events: data appropriateness, readiness, reliability, sensitivity and sufficiency. In what follows, we link our proposed dimensions to existing terminology in Wang and Strong (1996) and Schlegel et al. (2020) where possible, and mention other related research in the field.

**Appropriateness:** This dimension considers the question "Is the data relevant to the data science questions posed (i.e., problem-specific)?", and closely aligns with "relevancy" (Wang and Strong, 1996; Schlegel et al., 2020). Examples of context-specific uses of relevancy include Daly (2006), which provide a comprehensive set of guidelines for analysing specific climate datasets. In their guidelines, the authors outline the relevancy of specific climate factors for different types of analysis, and Schlegel et al. (2020) provides domain-specific concepts to assess data suitability along the production chain for defect prediction.

**Readiness:** This dimension considers the questions "Is the data analysis-ready?", and includes "completeness" and "accessibility" (Wang and Strong, 1996; Schlegel et al., 2020). Technology readiness levels (Dunbar, 2017; Hirshorn et al., 2017) was developed in the 1970s as a standardized technology maturity assessment tool in the aerospace engineering sector. NASA developed a scale with multiple degrees which has increased from 7, in the early years (Sadin et al., 1989), up to 15 degrees in the latest reviews (Olechowski et al., 2020).

Similarly, Lawrence (2017) proposed the use of data readiness levels, giving a rough outline of the stages of data preparedness and speculating on how formalization of these levels into a common language could facilitate project management. Building on this previous work, we specify four data readiness levels but specifically within the context of datathons challenges.

Afzal et al. (2021) outlines a recommended task-agnostic 'Data Readiness Report' template for a project's lifecycle, which includes different stakeholders involved such as final contributions by data scientists. Castelijns et al. (2020) identifies 3 levels of data readiness and introduces `PyWash`, which is a semi-automatic data cleaning and processing `Python` package. Their levels also include a series of steps which includes feature processing and cleaning. In their works, the use of 'data readiness' refers to the end-to-end lifecycle of a data science project, including steps typically taken during a datathon and after. Other works focus specifically on 'big data' readiness and in country-specific ecosystems which may not be easily extendable (Austin, 2018; Joubert et al., 2023; Hu et al., 2016; Zijlstra et al., 2017; Romijn, 2014). Since datathons are not confined to a single use-case, we believe it would be more appropriate to create a framework that can work across different applications.

**Reliability:** This dimension considers the question "How biased is the data?", and includes "objectivity" (Wang and Strong, 1996). The reliability of data is crucial for AI applications as it directly impacts the accuracy and fairness of the outcomes. Biased data which does not represent the problem fairly returns unreliable results (Ntoutsi et al., 2020). Drawing from established frameworks such as Suresh and Guttag (2019) and Ntoutsi et al. (2020), we identify several potential sources of bias that can undermine the reliability of the data. Selected examples may include: *(i)* historical bias, which arises from past societal or systemic inequalities; *(ii)* representation bias, where certain groups or perspectives are underrepresented in the data; *(iii)* measurement bias, which stems from flawed or subjective data collection methods; and *(iv)* aggregation bias, which occurs when data is combined or summarised in a way that does not accurately represent each sub-group within the wider population.

**Sensitivity:** This dimension considers the questions "How sensitive is the data?". We build on existing work from Arenas et al. (2019) by incorporating their data sensitivity analysis categorisation to our framework. They propose a defined tier-based system for sensitive data management in the cloud, which contain: *tier 0*, datasets with no personal information or data which carries legal, reputational, or political risks; *tier 1*, data that can be analysed on personal devices or data that can be published in

the future without significant risks, such as anonymised or synthetic information; *tier 2*, data that is not linked directly to personal information, but may derive from it through synthetic generation or pseudonymisation; *tier 3*, data that presents a risk to personal safety, health, or security on disclosure. This may include pseudonymous or generated data with weak confidence in the anonymisation mechanism or commercially and legally confidential material; *tier 4*, personal information that poses a substantial threat to personal safety, commercial, or national security and is likely to be subject to attack.

The data used in the study may also need to adhere to various regulations, including but not limited to the Data Protection Act (Government of the United Kingdom, 2018), GDPR (European Commission, 2016), and specific regulations governing special categories of data (Information Commissioner's Office (ICO), 2021; European Parliament, 2016). Additionally, different countries or sectors may have their own specific regulations in place. It is important to note that the application of these laws is evaluated on a case-by-case basis, and this paper does not provide specific guidance on how to comply with them.

**Sufficiency:** This dimension considers the question "How much data is enough data for a given datathon?", and includes "appropriate amount" (Wang and Strong, 1996; Schlegel et al., 2020). This question is highly problem and method-specific, and has been formulated in many research fields. In an early study, Lauer (1995) explored a series of issues in analysing data requirements for statistical Natural Language Processing (NLP) systems and suggested the first steps toward a theory of data requirements. Questions such as (i) " "Given a limited amount of training data, which methods are more likely to be accurate?", or (ii) "Given a particular method, when will acquiring further data stop to improve predictive accuracy?", were addressed. In intelligent vehicles, Wang et al. (2017) proposed a general method to determine the appropriate data model driver behavior from a statistical perspective. They point out that this question has not been fully explored in their field of research, but it is highly relevant given that insufficient data leads to inaccurate models, while excessive data can result in high costs and a waste of resources. Cho et al. (2015) presented a general methodology for determining the size of the training dataset necessary to achieve a certain classification accuracy in medical image deep learning systems. They employed the *learning curve* method (Figueroa et al., 2012), which models classification performance as a function of the training sample size. They pointed out that, in medical image deep learning, data scarcity is a relevant problem. Developing a methodology to estimate how large a training sample needs to be in order to achieve a target accuracy is therefore highly relevant. Since datathons include a wide range of applications and approaches, we adopt a different take on "sufficiency" and propose general guidelines based on experiences in prior datathons. In our case studies, we make case-specific observations (see section 5).

## 3    Data Study Groups

Data Study Groups (DSGs) are an award-winning[1] collaborative datathon event organised by The Alan Turing Institute, the UK's national institute for data science and artificial intelligence. Each DSGs consist of multiple datathons (or "challenges"), typically 3-4, that are worked on collaboratively by a single team (rather than multiple teams competing with each other). The aim of DSGs is to provide opportunities for organisations and participants from academia and industry to work together to solve real-world challenges using data science and ML methodologies. The DSGs are managed and prepared by a specialised internal team of event organisers and interdisciplinary academic support staff.

The events typically run for a week, during which 8-12 participants work in teams to explore and investigate the challenges. The events are overseen by experienced researchers that act as "principal investigators" (PI) and "challenge owners" as subject-matter experts. Both provide technical guidance and support for the teams during the week. DSGs have tackled various challenges across different sectors, such as healthcare, finance, and transportation. Figure 1 shows the breadth of challenges undertaken, with $\geq 69$ challenge owners spanning $\geq 77$ datathons across different domains. Note that we have carried out over 80 datathons, but not all datathons have available meta-data for analysis.

During the datathon, a technical report is written by the participants, and edited by the PI after the event. The report collates the work that was carried out by the datathon team and articulates successes, challenges, and recommendations for future analysis directions. The report is not intended to undergo

---

[1]https://www.praxisauril.org.uk/news-policy/blogs/turing-data-study-groups-ke-award-winners

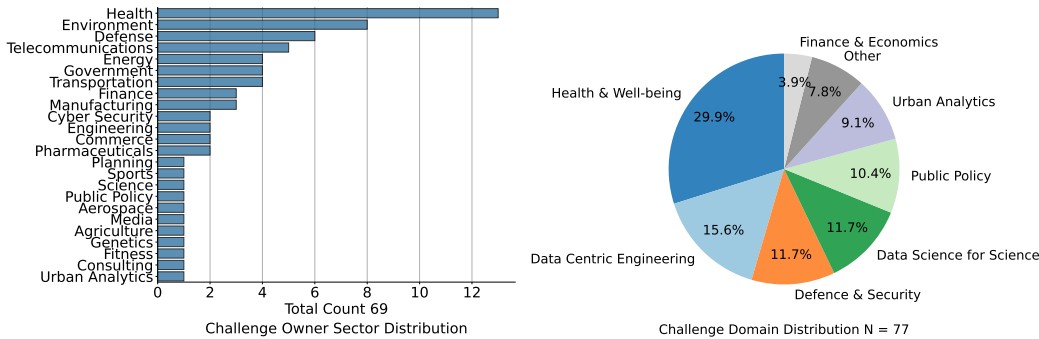

Figure 1: Left: Bar chart showing the distribution of sectors from the challenge owners ($n = 69$). Right: Distribution of datathon topics ($N = 77$) for the Data Study Groups (DSGs).

a traditional academic peer-review process to assess novelty. Rather, members of the DSG team at the Alan Turing Institute review the report for scientific rigour and clarity before publication.

The internal peer-review process is aligned with considerations of responsible research and innovation. DSG challenges hosted at the Turing are all assessed for ethical considerations following guidance published by the Public Policy team (Leslie, 2019), therefore, ethics is not an additional data dimension in our process, it is a separate step that is comparable in size to data assessment. Although we recommend that all datathon projects be assessed for data protection and ethical considerations, these processes are outside this paper's scope, as it varies by country, organising institution and partners needing for an in-depth discussion even within fields to be useful. Our data assessment framework complements institutional policies and can fit within data governance laws across different countries.

## 4 Data Assessment Matrix

| | **Insufficient** | **Developing** | **Functional** | **Optimal** |
|---|---|---|---|---|
| **Appropriateness** | Data unrelated to the research question(s). | Target variables are inappropriate for the research question. | Features aren't entirely appropriate for the research question. | Clear relation between features and target variables. |
| **Readiness** | No data collected or collection methodology. | Some collected data or clear collection methodology in place. | Data collected, but requires some merging and unifying. | Data collected and cleaned. |
| **Reliability** | Strongly biased data. | Indeterminate source of bias. | Determinate source of bias which can be accounted for. | Weak or clear of bias. |
| **Sensitivity** | Highly risky sensitive data. | Requires extensive security measures. | Data does not present significant risk but may be confidential. | Limited risks associated with the data, usually publicly available data. |
| **Sufficiency** | Prior experience shows low success rate. | Does not meet the level of previous succesful events. | Previous events have successfully operated at this level. | Extensive data in excess of previous successful events. |

Figure 2: Summary data assessment matrix for the five data dimensions proposed in this paper.

### 4.1 Appropriateness: Is the data relevant to the questions posed?

Data appropriateness refers to how relevant the data is for the challenge or research questions at hand:

*Insufficient:* Data is not related to the challenge question or is completely unrelated to the problem at hand. As an example, a datathon team is tasked with predicting consumer behavior for a new product launch but is given data on historical weather patterns instead.

*Developing:* Data is related to the challenge question, but the target variables needed to answer the research questions are missing. For example, a datathon may seek to predict stock price movements

from historical transaction data but without stock prices, it would be challenging to produce useful results.

*Functional:* Data is appropriate for the challenge but the features can notably be enhanced with data the partner organisation could provide. For example, a datathon team is given a dataset on website traffic and user behavior, which is relevant to their challenge, but they identify that additional data on customer feedback would be helpful in making more accurate predictions.

*Optimal:* Data is highly appropriate and relevant to the challenge. A datathon team is given comprehensive datasets which provide all the necessary information for making accurate predictions and developing effective solutions. Appropriate meta-data and collection processes are also provided.

## 4.2 Readiness: Is the data analysis ready?

Data readiness refers to the degree to which data is prepared and available in a datathon. This includes factors such as the quality and completeness of the data, the availability of relevant metadata, and the ease with which the data can be accessed and analysed. Ease and effectiveness of analysis are closely coupled to effective and thorough data documentation Gebru et al. (2021); Holland et al. (2018), which increase transparency, understandability, and accountability of data and model usage (Mitchell et al., 2019). Data readiness categories are as follows:

*Insufficient:* No data has been collected yet, and there is no methodology for doing so. For example, if the datathon involves collecting data from sensors, and there isn't a clear view of where the data is being stored or what it actually contains, the data readiness degree would be *insufficient*.

*Developing:* Some data has been collected, and a firm methodology is in place for doing so. However, there may still be some uncertainty around its quality and completeness. For example, if the datathon involves analysing customer data from a company, and even though they have access to their data, parts of it are missing, or inconsistent, the data readiness degree would be *developing*.

*Functional:* Data is collected, but some merging and unifying still need to be done during the event. The data may require some cleaning and preprocessing, but with minor processing can serve to answer the challenge questions. For example, if the datathon involves analysing public transportation data from different sources, and some data needs to be merged, aggregated, and formatted to fit into the analysis, the data readiness degree would be *functional*.

*Optimal:* Data is collated, clean, with no missing gaps, and can potentially be fed to a learning algorithm with minimal processing. The data is also documented with the data dictionary and data cards, such as the ones highlighted in previous research (Gebru et al., 2021; Pushkarna et al., 2022) but in the context of datathons. It is worth noting that dataset documentation in general remains a developing field of study, and datathon-specific recommendations or guidelines are yet to be established. For example, if the datathon involves analysing a well-established dataset, such as the MNIST dataset, the data readiness degree would be *optimal*.

## 4.3 Reliability: How biased is the data?

Reliability here refers to the extent to which the data accurately represents the population or phenomenon without systematic data collection errors or distortions.

*Insufficient:* Data has significant known biases, and conclusions drawn from the analysis will be undermined. For example, in a datathon studying democratic voting patterns, the dataset only includes responses from a small, self-selected group of participants, rendering it insufficient to make accurate inferences about the broader population's voting behavior.

*Developing:* Data has unknown or indeterminate roots of bias, and hence no firm conclusions can be made using only these data sources. For example, in a datathon analysing economic indicators of different countries, it is suspected that there are geographical and temporal inconsistencies in the data collection. As a result, the dataset is considered to be in the *developing stage* of reliability.

*Functional:* Data has known biases that will impact the analysis and can be corrected or accounted for as a work limitation. For example, in a datathon analysing customer feedback surveys, it is known that the surveys were only completed by customers who voluntarily provided feedback, introducing

a self-selection bias. The organisers acknowledge this limitation and proceed with the analysis, highlighting the potential impact of the bias on the conclusions.

*Optimal:* Data does not suffer from a known source of biases with significance for the conclusions. For example, in a datathon exploring climate change impacts, the dataset comprises meticulously collected climate measurements from a comprehensive network of sensors across multiple regions.

### 4.4 Sensitivity: Is the data private or confidential?

Data sensitivity refers to the level of confidentiality and privacy associated with specific types of data. It indicates the degree to which the data is considered private, personal, or valuable, requiring enhanced protection. Ensuring these safeguards may increase the difficulty of hosting a datathon. Therefore, we map the sensitivity tiers described by Arenas et al. (2019) to our analysis framework.

*Insufficient (Tier 4):* Highly sensitive data presents significant threats, making hosting a datathon impractical. Due to potential legal and personal risks, datathons are not supported at this tier. For instance, handling datasets with commercial or national security sensitivities. This aligns with the UK government's "*secret*" classification (UK Cabinet Office, 2018). At Tier 4, the risk of malicious actors infiltrating the datathon team becomes notable.

*Developing (Tier 3):* Careful consideration and extensive security measures would be required prior to hosting a datathon in this tier. Datathon participants would be restricted in their use of tools, location and network connections, increasing system friction. For example, in a medical context, a dataset containing highly sensitive patient health records, including names, addresses, and detailed medical histories, poses a substantial threat

*Functional (Tier 2):* Data in this tier does not present significant personal, legal or commercial risk. However, sufficient measures must be taken to mitigate processing risks, such as *(i)* data de-anonymisation, *(ii)* competitive data leakage *(iii)* or inadvertent disclosure. These measures increase the difficulty of participants interacting with the data. For example, an educational dataset with anonymized student records that still hold the risk of re-identification. Hosting a datathon using this data would necessitate robust security measures to prevent any potential breach of student privacy and unauthorized access to sensitive academic information.

*Optimal (Tier 0/1):* Since risks associated with this data are limited, hosting datathons in virtual environments is usually unnecessary. Free data access to the participants reduces system friction. A dataset containing weather records from a public weather station that doesn't include any personal or sensitive information would fall under this tier.

More details, including a discussion on the classification process, as well as the technical requirements to build such environments can be found in (Arenas et al., 2019).

### 4.5 Sufficiency: How much data is enough data for a given datathon?

Data sufficiency refers to the quantity of available data to investigate the challenge questions given a proposed approach. In addition to case-by-case (e.g., depending on the approach and the question) considerations, we propose below an assessment based on previous experiences of similar challenges. In section 5, which is problem-specific, we make case-specific comments on data sufficiency.

*Insufficient*: Prior experiences have shown that working with similar data quantities on projects with similar objectives hinders the overall success and outcomes of the datathon. For example, in a medical research datathon aiming to predict patient outcomes, the provided dataset does not include information about clinical trials.

*Developing*: Prior experience has shown that the quantity of data available is not substantial enough to address the research question comprehensively. This quantity of data raises questions about its ability to offer statistical evidence and comprehensive insights into the challenge questions, unlike the *insufficient* category, data may be present but not adequate for comprehensive insights. For example, a dataset containing a few time-series signals in an industrial failure detection manufacturing datathon not allowing to distinguish between normal and failure modes (Bocincova et al., 2022).

*Functional*: Although not surpassing previous successful datathons, prior experience has shown that the quantity of data is sufficient to provide useful insights and statistical evidence into the challenge

questions. Any insufficiency can be accounted for as a work limitation and discussed in future work. For example, in a datathon focused on developing an automated system for classifying sea ice and mapping seals to aid ecological monitoring, the geospatial data provided covered only two focal areas, which limits the potential generalizability of the developed system to broader regions (Acharya et al., 2020).

*Optimal*: This category represents an amount of data which surpasses previous successful datathons. The dataset(s) is extensive (potentially more than what was proposed initially), offering a rich source of information for participants to explore and leverage as they tackle the challenge questions. For example, in a datathon that aims to predict the functional relationship between DNA sequence and the epigenetic state, having access to comprehensive and well-labelled genomic datasets (Phuycharoen et al., 2022).

## 5 Case Studies

We evaluated the quality of data in past DSGs by referring to the final reports available on The Alan Turing Institute's website[2] and show an aggregate summary of the results in Figure 3. It is important to acknowledge a publication bias in these reports, as only those that meet minimum quality criteria determined by the reviewing team are made available. Moreover, the reports from early DSGs might not have been published due to problems ranging from unresolved sensitivity issues to misaligned expectations between organisers and challenge owners. Consequently, our focus was primarily on the later versions of the DSG.

This section presents the ten most recent datathons case studies, selected from over 80 datathons, in Figure 3 we provide a summary of the results of the data assessment of the case studies. These case studies have data, challenge descriptions, and limitations extracted from the publicly available DSG Reports. They are divided between the main body of the report and the appendix.

|  | Insufficient | Developing | Functional | Optimal |
|---|---|---|---|---|
| Appropriate | 0 | 2 | 3 | 5 |
| Ready | 0 | 2 | 5 | 3 |
| Reliability | 0 | 2 | 3 | 5 |
| Sensitivity | 0 | 0 | 6 | 4 |
| Sufficiency | 0 | 4 | 3 | 3 |

Figure 3: Report count data assessment classification of the last 10 DSG reports

### 5.1 Cefas: Centre for Environment, Fisheries and Aquaculture Science

The datathon aimed to address the challenge of automating the classification of plankton species and detritus in millions of images collected by the Cefas Endeavour research vessel's Plankton Imager system, enabling efficient analysis and contributing to the understanding and conservation of marine ecosystems (Asthana et al., 2022).

---

[2]https://www.turing.ac.uk/collaborate-turing/data-study-groups

**Appropriateness.** *(Optimal)* Plankton images and their corresponding labels were present in the data: "expert manual classification (labels) allowed challenge participants to verify the accuracy of the automated classification methods explored."

**Readiness.** *(Functional)* Images had faulty labels: "[. . . ] we discovered that a number of the images labelled as detritus were in fact empty."

**Reliability.** *(Optimal)* When writing the challenge proposal, we identified a potential bias in the data collection method: all images were collected by the same imager system on the same vessel. However, Cefas did not assess this risk as substantial, which was corroborated by the project outcome.

**Sensitivity.** *(Optimal)* The data is open access, collected by the Plankton Imager[3].

**Sufficiency.** *(Functional)* Some species of plankton were under-represented, making it an imbalanced problem which needed to be accounted for in the analysis (e.g. using data augmentation).

## 5.2 The University of Sheffield Advanced Manufacturing Research Centre: Multi-sensor based Intelligent Machining Process Monitoring

This challenge aimed to enhance process monitoring in modern manufacturing using machine learning techniques to analyse multiple sensor measurements (Bocincova et al., 2022). Two datasets were provided for this challenge containing time-series process signals of the machine operating in both normal and failure modes. The datasets were recorded during separate machining trials.

**Appropriateness.** *(Developing)* The target variables needed for the challenge were not present in the data: "the provided dataset does not contain any data describing the transition period between the normal operation mode and failure modes"

**Readiness.** *(Functional)* The data requires aggregation before being used: "datasets were collected at extremely high sampling frequencies . . . resulted in excessively big datasets"

**Reliability.** *(Functional)* The data experiments were not recorded in the same instance so although measures were taken to ensure that they were as identical as possible, there was a known bias that could be accounted for: "The two provided datasets were recorded during two separate machining trials [...]. However, a number of identical components were machined in each trial to act as repeats for the dataset."

**Sensitivity.** *(Functional)* The data was commercially sensitive and was mapped as Tier 2.

**Sufficiency.** *(Developing)* Not enough data for the analysis: "There were still not enough multivariate time series produced by different runs of the experiment to accurately use image classification."

## 5.3 CityMaaS: Making Travel for People in Cities Accessible through Prediction and Personalisation

The aim of this datathon was to develop a platform that provides reliable information on the accessibility of destinations (Aragones et al., 2021). In particular, participants were tasked to $(i)$ predict the accessibility of points of interest (POIs) in a city; $(ii)$ to develop a personalised route-planning algorithm with accessibility constraints and for different "persona" types of their users. For $(i)$, CityMaaS provided manually enhanced point data from OpenStreetMap (OSM) (OpenStreetMap contributors, 2021) and for $(ii)$ information on (synthetic) potential end-user personas were provided (i.e. place of residence, civil status, profession, income, conditions, etc.).

**Appropriateness.** *(Developing)* The enhanced OSM dataset provided for the task $(i)$ was appropriate for predicting POI accessibility. However, for task $(ii)$, participants were only given information about the profile of potential users.

**Readiness.** *(Functional)* Significant data cleaning was necessary before the analysis was performed; for example, participants noted the need to "homogenise" the POI data to avoid naming errors in the dataset.

---

[3]https://data.cefas.co.uk/view/20507

**Reliability.** *(Developing)* The dataset for POI prediction had a class imbalance with 73% of POIs being labelled as accessible. POI data was taken from OSM where users are allowed to input tags leading to possible errors and annotation bias.

**Sensitivity.** *(Functional)* There was commercial sensitivity for CityMaaS in providing the enhanced OSM dataset provided and the (synthetic) user profiles.

**Sufficiency.** *(Developing)* In particular, for the routing task, no routing data was provided. Participants searched for additional open-source data sources to supplement the data provided by CityMaaS.

# 6 Recommendations

Recommendations for effectively working with data in datathons can be organised based on the preliminary stages of the event. The following suggestions draw from our investigation of the case studies and are supported by learning from previous datathons. Given that our paper aims to suggest a small number of dimensions and recommendations that can be applied to a broad range of datathons, the recommendations below are general and broad-purpose. Specific recommendations for moving from one tier to another or on what constitutes "good enough" will need to be assessed case-by-case, as illustrated in section 5.

**Preparatory phase:** Prior to the datathon, one important recommendation is for the event organisers to engage the challenge owners actively. Our framework can help focus discussions with challenge owners on dimensions where scores are below "functional". Their understanding of the problem and domain expertise can greatly improve the data, and early collaboration helps align the objectives and expectations on both sides, increasing the likelihood of a fruitful event. For example, in our Strathclyde challenge (Amaizu et al., 2022), the initial proposal was misaligned in terms of readiness, and the project focus was shifted from the participants executing simulations at the datathon to the challenge owners providing them so that the participants could focus on pattern-finding. Conversely, the Cefas challenge described in section 5.1 achieved greater success due to significant involvement from the challenge owners.

**Refinement phase:** As the datathon approaches, it is beneficial to do sanity checks on data readiness and consider changing the challenge questions based on input from the PI. Again, our framework can help focus the efforts of PIs on dimensions where scores are below "functional", and they can consider the viability of the datathon on a case-by-case basis. This phase can help to refine the problem formulation and avoid appropriateness issues, ensuring alignment with the goals of the datathon. For example, by earlier involvement of a PI, we could have detected the lack of appropriate data or the insufficiency of the multivariate series in the Sheffield challenge in section 5.2.

**At the datathon:** During the datathon itself, the PI will be present to help guide participants, having considered and reflected on any data issues and mitigants prior to the datathon. It is also essential to leverage the expertise and feedback of the participating data scientists. Their insights and perspectives can serve as valuable expert advice in identifying areas for improvement across the data quality dimensions. By actively seeking and incorporating participant feedback, organisers and participants can make small adjustments that enhance data utility, facilitating more effective and impactful analyses during and in future events. For example, in the CityMaas challenge of section 5.3, participants found data reliability issues (inconsistent labelling by users), or in Get Bristol Moving (Mougan et al., 2020) alternative sources of open-source data were found that allowed decreasing the sensitivity tier.

# 7 Conclusions

In this paper, we have analysed data in the context of datathons along five key dimensions: *appropriateness*, *readiness*, *reliability*, *sensitivity* and *sufficiency*. We then mapped 10 case studies from Data Study Groups to our framework and provided a set of recommendations based on the lessons learned. By doing so, we hope to improve the handling of data for organisations prior to datathon events.

Our proposed qualitative analysis provides a degree of data status across several perspectives; these degrees can be adapted or extended, similar to the Technology Readiness Levels provided by NASA (Sadin et al., 1989), which have been extended through time and further work.

## Acknowledgements

The DSG team wish to thank the past contributions of Dr Merve Alanyali, Dr Alex Bird, Rebecca Blower-Harris, Dr Ayman Boustati, Xander Brouwer, Dr Zhenzheng Hu, Dr Chanuki Illushka Seresinhe, Catherine Lawrence, Dr Zhangdaihong Liu, Dr Bilal Mateen, Frank Murphy, Daisy Parry, Katrina Payne, and Dr Mahlet Zimeta. With special thanks to Prof Sebastian Vollmer for bringing the concept to us and starting DSGs at the Turing back in 2016, and Dr Franz Kiraly for his work and input in the early years. We also wish to thank all past Challenge Owners, DSG PIs and participants for their invaluable contribution and for making DSGs possible.

Carlos Mougan was funded by the European Union's Horizon 2020 research and innovation program under the Marie Skłodowska-Curie Actions (grant agreement number 860630) for the project: "NoBIAS - Artificial Intelligence without Bias".

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

# A  Appendix: Case Studies

## A.1  WWF: Smart Monitoring for Conservation Areas

The datathon focused on developing a NLP system to automatically detect news articles reporting emerging threats to protected areas, enabling real-time monitoring and proactive conservation efforts by WWF and the wider conservation community (Hosseini et al., 2020). The dataset contained approximately 45,000 files describing news stories relevant to UNESCO World Heritage Sites, of which 135 were labeled as 0-'No threat', 1-'Threat detected but not actors' and 2-'Threat with actors'.

**Appropriateness.** *(Optimal)* The data was obtained using the Google News API and was manually labeled by WWF experts to be relevant to the challenge.

**Readiness.** *(Developing)* Even thought there was a data collection methodology in place, the team had to increase the number of data points available during the event: "...this dataset was expanded during the DSG week to 1,000 labelled data points".

**Reliability.** *(Functional)* Data was manually labelled by WWF experts and scraped using the Google News API, creating a before-hand known data collection bias.

**Sensitivity.** *(Optimal)* News articles were scraped using Google News API which made it an open source project of Tier 0.

**Sufficiency.** *(Developing)* The number of news scraped was potentially sufficient, but the amount of labeled data was insufficient: "...initially a training dataset of 135 news articles annotated by the WWF experts were supplied".

## A.2  British Antarctic Survey: Seals from Space

The datathon aimed to create an automated system for classifying sea ice, mapping seals, and exploring environmental factors to facilitate ecological monitoring and understanding of the Antarctic ecosystem (Acharya et al., 2020). The data consisted of GeoTIFF files covering two focal areas, the Antarctic Peninsula (Crystal Sound and Marguerite Bay) and Signy Island. Location data from over 2000 manually-counted seals accompanied these images.

**Appropriateness.** *(Functional)* Even though the images and seal location data were provided, some important covariates such as nursing locations, ice features, or marine food availability, were not provided.

**Readiness.** *(Optimal)* Data was in an appropriate state to apply learning algorithms.

**Reliability.** *(Functional)* Data collection bias since seal counting has only been attempted on some regions: "as we do not have image scenes of locations in which counting has been attempted but zero seals have been observed, we cannot generalise our results to other regions of ice".

**Sensitivity.** *(Optimal)* The data was open source and classified at Tier 1.

**Sufficiency.** *(Functional)* The variability on spatial points was restricted to two locations representing typical seal communities. In order to properly validate the model, it should be done by statistical testing in a new region outside of the given ones.

## A.3  DWP: Department for Work and Pension

The UK Department of Work and Pensions (DWP) processes a large amount of personal data, including demographics, family circumstances, and financial details, which presents opportunities for large-scale data analysis and concomitant risks to personal privacy. This datathon (Deineha et al., 2021) evaluated the potential for generating synthetic instances from the original data while preserving utility. The DWP provided two datasets of Universal Credit claimant data. Both datasets were synthetic data, representing the original sensitive population, and are close in structure to existing DWP datasets.

**Appropriateness.** *(Optimal)* Since the challenge regarding the evaluation of the similarity of generated datasets, the provided target data and comparator-generated sets were highly appropriate to this task.

**Readiness.** *(Functional)* Additional cleaning and merging of the target datasets was necessary before processing; secondary datasets required alignment work.

**Reliability.** *(Optimal)* Sources of bias in the data collection methodology were identified due to the generation mechanisms for the target dataset. However, these were limited in their effect on the evaluation metrics used.

**Sensitivity.** *(Functional)* While the target data did not contain real personal identifiers, it was still considered confidential and a potential vector for data breaches.

**Sufficiency.** *(Optimal)* A large number of available target data rows exceeded the scope of previous challenges in synthetic data generation.

## A.4 Dementia Research Institute and DEMON Network: Predicting Functional Relationship between DNA Sequence and the Epigenetic State

This datathon aimed to improve regulatory genomic predictions to understand the impact of genetic variants on gene expression and regulation in disease-relevant cell types, with a focus on dementias such as Alzheimer's disease, and potential implications for drug target development.(Phuycharoen et al., 2022). Four different datasets were provided, containing information about the DNA sequence and the associated chromatin states.

**Appropriateness.** *(Optimal)* The data provided contained the features and labels of interest. For example, "...the outputs are always trained to predict one of two epigenetic signals ...in 3 cell types".

**Readiness.** *(Functional)* Data required merging and aggregating between datasets to be able to extract insights "Most of the samples were provided with associated DNase-seq data [...] some cell lines [...] were provided with ATAC-seq data instead. Although both assays are broadly used to"

**Reliability.** *(Functional)* The datasets used were open-source and annotated by domain experts. Some examples of the datasets used were: EpiMap[4] and Human Genome 38[5] There were some gaps in the data which the authors were not able to account for. For example, "All profile data ...included occasional not a number (NaN) values."

**Sensitivity.** *(Optimal)* The datasets used were open-source and was mapped to tier 0.

**Sufficiency.** *(Optimal)* 4 well-labelled and large genomic datasets were used for the challenge.

## A.5 Automating Perfusion Assessment of Sublingual Microcirculation in Critical Illness

This challenge sought to determine if a validated measure of microcirculatory perfusion, specifically the microcirculatory flow index, can be directly predicted from a video sequence obtained through darkfield microscopy. The goal is to enable automatic, real-time analysis of the videos, reducing the labour-intensive manual analysis process and facilitating the integration of microcirculatory targets into clinical trials for better patient outcomes (Tomlinson et al., 2022). The data comprised 800 grayscale videos of different lengths and the quality obtained via DFM from 52 patients monitored over four days

**Appropriateness.** *(Optimal)* The obtained data and labeled was specifically designed for the challenge. "These videos have been manually analysed to obtain perfusion parameters per short clip, labelling each data point..."

**Readiness.** *(Functional)* The dataset was ready but required minor renaming, merging and feature extraction.

**Reliability.** *(Functional)* There were cases in which a video had no perfusion parameters and vice versa. "some mismatches were identified ...cases in which a video had no perfusion parameters and vice versa".

**Sensitivity.** *(Functional)* The data was commercially sensitive due to containing videos from patients and was mapped to a Tier 2.

---

[4] http://compbio.mit.edu/epimap/
[5] https://www.ncbi.nlm.nih.gov/assembly/GCF_000001405.39

**Sufficiency.** *(Developing)* Since the data relied on manual labeling, the total amount of data was not sufficient to provide statistical validity for the method."[. . . ]dataset is comprised of a smaller number[. . . ]"

## A.6   Entale: Recommendation Systems for Podcast Discovery

The datathon aimed to develop methods for capturing relationships between podcasts, incorporating information about the content discussed within them and using this information to produce podcast recommendations (Chan et al., 2021). The data provided included user data (episode and show listened by the users) and podcast data (episode transcripts and show descriptions).

**Appropriateness.** *(Optimal)* Since the challenge was to analyse podcast episode content and to produce episode recommendations, user history and podcast data (episode transcripts, episode and show meta-data) is highly appropriate.

**Readiness.** *(Functional)* Merging and cleaning of the two datasets was necessary before analysis.

**Reliability.** *(Functional)* Bias was induced by a small overlap between datasets, and since the participants were tasked with producing a recommender system, they had some measurement bias when evaluating their methodology since they should be evaluated with new data.

**Sensitivity.** *(Functional)* Data was provided by Entale, and so there was a commercial risk for the data.

**Sufficiency.** *(Functional)* The two datasets provided were large and enabled users to tackle the research question, but the overlap between user episodes listened to, and episode transcripts was only 10% making it challenging to incorporate user data with detailed episode content fully.

## A.7   Odin Vision: Exploring AI Supported Decision Making for Early-Stage Diagnosis of Colorectal Cancer

Odin Vision is a UK company that has developed AI technology for detecting and characterising bowel cancer. The aim of this datathon was to explore methods that enhance the explainability of Odin-Vision's current machine learning models to aid clinical decision-making by (i) adding a measure of predictive uncertainty along with class prediction, (ii) making the classifications more understandable to the clinicians, and (iii) exploring methods that can automatically learn representations of features using generative models (Almarzouq et al., 2021). The provided data collated three public datasets of colorectal polyp images, obtained by merging data from white light endoscopy and narrow-band-imaging light sources.

**Appropriateness.** *(Functional)* A total of three image datasets were employed. Two of them were publicly available and contained both labelled and unlabelled images. The third one was a high-quality, non-public dataset that was not made available to the datathon participants but was employed by Odin Vision to train the models developed during the datathon.

**Readiness.** *(Developing)* Many of the images from the publicly available datasets were of low quality due to blur and the presence of reflective spots. A considerable number of images had to be either removed or pre-processed in order to alleviate this issue.

**Reliability.** *(Developing)* In addition to the positive-case bias resulting from unbalanced classes (see section on Sufficiency), the datathon participants concluded that the best approach was to train the models predominantly on Odin Vision's higher quality, non-public dataset, and to use a test set from the publicly available images. As the train and test sets were not drawn from the same distribution, the reported quantitative results might be affected by the distributional shift between the train and test datasets.

**Sensitivity.** *(Optimal)* The datathon participants only had access to open-source datasets and not to Odin Vision's non-public data, which made this a Tier 0 challenge.

**Sufficiency.** *(Functional)* Due to the nature of the datasets (real-world medical images), these were relatively small, with even fewer positive labelled data, which resulted in an insufficient amount of data for learning algorithms and drawing statistically significant answers.

