# OpenReview forum: "How to Data in Datathons"
_NeurIPS.cc/2023/Track/Datasets_and_Benchmarks — NeurIPS 2023 Datasets and Benchmarks Poster_

### Official Review · Reviewer_oVgp · 2023-07-10
**Review for submission 197**

**Rating:** 6
**Confidence:** 4
**Clarity:** The paper is easy to read and follow.

**Strengths:**

The literature review is very well written and clear, and covers a large range of relevant fields.
While the proposed method may only be relevant to limited communities, the dimensions regarding data quality evaluation could spark the interest of a much larger community.

**Additional Feedback:**

See the above opportunities for improvement.

**Correctness:**

There are no concerns regarding the correctness of the submission. There are, however, several concerns regarding the data assessment matrix and the evaluation of the five data dimensions.

**Documentation:**

-

**Ethics:**

-

**Limitations:**

Some limitations of the work are mentioned, but they should be discussed in more detail and their impact should be made more clear.

**Opportunities For Improvement:**

As mentioned before, the related work section is very well written. However, when reading the description of the 5 dimensions identified, it seems like just one is adapted from the literature. Since there are so many dimensions out there, it might make sense to reuse or adapt them as much as possible.
There is also little motivation for the scale of insufficient, developing, functional, and optimal. How were these values determined? why are these the most suitable?
It is very helpful to see examples for each scale and value pairing. However, it might be more useful to have the same running example for the entire section 4, and give some hypothetical examples for the relevant measures and values.
Tables and Figures are not referenced too well.
In general, it is unclear how the assessment matrix should be used and how the degree of readiness, for instance, can be evaluated. This is particularly difficult for cases where data quality is in general not of the best quality, amount, etc. Since some of these measures could be highly subjective, I would encourage the authors to make a more precise checklist that researchers can use in order to evaluate their datasets.
Ethical and privacy concern are always of high importance and relevance. These aspects, however, are not addressed in the current submission.

**Relation To Prior Work:**

The literature review is very well written and clear, and covers a large range of relevant fields.

**Summary And Contributions:**

The paper proposes 5 key dimensions (i.e., appropriateness, readiness, reliability, sensitivity, and sufficiency) for analysing data study groups. By looking at a broad range of such data study groups, the authors make recommendations for effectively working with data in datathons.

---

> ### Author Response · Authors · 2023-08-19
>
> Dear reviewer,
>
> Many thanks for the time invested in reading our paper and for your constructive feedback. We address the main comments in point form here:
>
> > How did we come up with the scaling criteria? Why are they most suitable?
>
> The scaling criteria were developed through a combination of hands-on experience, literature reviews, analysis of past datathons, and their application to real-world cases, ensuring their relevance and practicality.
>
> As we have stated in the last paragraph of the conclusions:
> “Our proposed qualitative analysis provides a degree of data status across several perspectives; these degrees can be adapted or extended, similar to the Technology Readiness Levels provided by NASA, which have been extended through time and further work.”
>
>
>
> > Why did we not consider adapting or reusing literature frameworks? It seems to be the only one that has been adapted from current literature.
>
> In our related work section, we were careful to adapt and reuse existing frameworks, where applicable. Of the five dimensions proposed, four (Appropriateness, Readiness, Reliability, and Sufficiency) consolidate existing data dimensions in the literature (as detailed in Section 2), and sensitivity builds on a previous existing project e co-author at the Alan Turing. Our goal was to suggest a minimalistic number of dimensions that can be applied to wide range of datathons.
>
> > Useful to provide examples of scaling and pairing (use cases).
>
> In Section 5, we have provided some examples based on prior Data Study Groups (DSG) to show how the framework can be used. Because we are using real-world cases and data, the challenges reflected indicate types of problems that occur when assessing the feasibility of a DSG. As noted in lines 284-285 in Section 5, the examples shown are only of DSGs which met the minimum criteria due to publication bias.
>
>
> > Clarity on how to use the framework and evaluate. Can we come up with a more precise checklist?
>
> As the reviewer has noted, some of these measures can be context-dependent, contingent on the specific problem being addressed and the research question at hand.
>
> Given the broad spectrum of subjects and scientific methods encompassed by datathons, our intent is to offer this framework as an initial guide for researchers involved in conducting datathons.
>
> Defining exact threshold and operationalising transitions between qualitative phases relies heavily on the project's unique parameters, which in turn are influenced by the researchers' prior expertise and methodologies, factors that are intricate and often challenging to quantify or generalize.
>
>
> > Paper does not address ethics or privacy (see below)
>
> While ethics is not discussed, privacy falls under section 4.4. L251 “Data sensitivity refers to the level of confidentiality and privacy”
>
> Ethics is not an additional data dimension in our process, it is a separate step that is comparable in size to data assessment (the focus of our paper). In particular, we conduct thorough and iterative ethics reviews that build on institutional guidance and guidance by Alan Turing's Public Policy programme. We have now clarified this in the paper and plan to write a follow-up focused solely on ethical considerations in datathons.

---

> > ### Comment · Reviewer_oVgp · 2023-08-28
> > **response to author rebuttal**
> >
> > Thank you for taking the time to answer to our concerns and improve some of these points in the submission. I believe the proposed framework will be a valuable resource for other researchers, so I'm inclined to raise my score.

---

### Official Review · Reviewer_ijbY · 2023-07-15
**Framework for Preparing Data for Hackathons, Lacking Actionable Recommendations**

**Rating:** 7
**Confidence:** 4
**Correctness:** No issues of correctness.
**Clarity:** No issues of clarity; the paper is we…

**Strengths:**

- Studying data hackathons is a useful avenue of work in which I haven’t come across much existing literature. In my experience, it seems like data hackathons would probably benefit from having some kind of standardization/framework to build from and fall back on when organizing.
- In particular, this paper seems to be (to the best of my knowledge) novel in that they focus on the pre-hackathon organization and preparation.
- It’s clear the authors have much experience in data hackathons. The case studies are interesting to read about.

**Additional Feedback:**

N/A

**Documentation:**

No documentation is necessary.

**Ethics:**

No ethical concerns.

**Limitations:**

There is not any discussion of potential negative societal impact in the paper. That being said, I do not personally see any negative societal impact for this work. It would be nice to see the authors provide some discussion of the limitations of this framework somewhere in the paper.

Also, notably, the NeurIPS paper checklist appears to be missing from this submission.

**Opportunities For Improvement:**

Major comments:
- As a reader, what I most want from this paper are recommendations that I could take with me when organizing my own hackathon. However, the recommendations provided in Section 6 are quite minimal and are not given in context of the five-part framework. For example, I’d want to know things like, “How can I assess my data sufficiency?”, “If I’ve determined that my data is at the Developing stage of reliability but is Optimal in terms of readiness, how should I proceed?”, “At what stage does my data need to be at before its ready for the hackathon? Can I use any data in a hackathon as long as I provide participants with the data’s scores in each of the five pieces of the framework?”. It is this kind of actionable information that I think is sorely missed in the paper. This would be a great opportunity for the authors to share their important data hackathon experiences with readers and provide concrete, actionable recommendations. These could then be demonstrated via the concreteness of the case studies.  Currently, however, the framework and case studies feel fairly disparate from the recommendations and guidance going forward.
- This paper is currently missing important discussion of: what makes preparing data for a hackathon different from preparing data for an analysis? What unique concerns does it invite? How does this work relate to recent data documentation initiatives in ML, such as Datasheets for Datasets? These are important issues for the paper to address tin order to clarify its goals and relationship to previous work.

Minor comments:
- Section 2.1: This section could be condensed into probably just one paragraph. This is some more space that could be dedicated to recommendations.
- Section 2.2: Not sure how this would be best addressed, but this section feels redundant when I get to reading Section 4. I understand that Section 2 provides the relation to prior work while Section defines the framework, but it felt a bit tedious to read about the five pieces of the framework, then read about DSGs, and then go back and recap and build upon the five pieces of the framework. It leads me to wonder if there is a better way to organize these sections so that each of the five elements is discussed once. This would be helpful to a reader revisiting the paper because they could e.g., find all the relevant info on “data readiness” in one section without having to go back and forth.
- Section 3 + Figure 1: This section could be condensed to focus only on what is necessary in order to convey the ideas of the data assessment framework. Though the information in this section and conveyed in Figure 1 is certainly interesting and demonstrates the breadth of DSGs, it is taking up valuable space. This is space that could be used to provide recommendations that readers who are hosting their own data hackathons (and aren’t involved with DSGs) could take from this paper. I suggest moving much of this material to the Supplement.
- Figure 1: I suggest switching the pie chart in Figure 1 to a bar chart. See Save the Pies for Dessert.
- Figure 3: This figure is not discussed anywhere in the main text. I suggest moving it to the Supplement.

**Relation To Prior Work:**

The paper makes it clear they are the first to focus on the data preparation “pre-hackathon”. However, the paper could benefit from making it more clear why applying the framework to data hackathons makes their work very different from Wang and Strong, 1996 and Schlegel et al., 2020, who they seem to borrow from quite heavily in their framework design. Again, I think it's important to address here: what unique challenges and concerns do data hackathons invite that this framework is more equipped to handle compared to previous frameworks?

As stated in “Opportunities for Improvement” above, the paper would also benefit from discussing recent data documentation initiatives in ML specifically, such as Datasheets for Datasets.

**Summary And Contributions:**

This paper proposes a five-part framework for assessing data that is to be used in a hackathon. The authors define and discuss levels of data: appropriateness, readiness, reliability, sensitivity, and sufficiency. They examine several case studies through the lens of this framework and drawing from these experiences, provide recommendations for future data hackathons.

---

> ### Author Response · Authors · 2023-08-19
>
> We appreciate the time and effort invested in evaluating our paper. Below, we respond to your comments and provide clarifications on the various aspects of the review.
>
> 1. Actionable Insights and Methodological Constraints:
>
> The reviewer highlights the absence of actionable insights on transitioning data from one stage to another within our framework. We acknowledge this limitation and would like to clarify that suggesting specific recommendations for such transitions, or for what constitutes “good enough”, is a complex and not a “one size fit all” task.  Furthermore, supporting such assertions would require a solid foundation in a rigorous methodology, a task of considerable complexity. We illustrate this in Section 5, where we have example case studies with “developing” across all categories except for sensitivity to give concrete examples of how a classification that falls below “functional” can manifest or be dealt with in different datathon problems.
>
> Our aim was to establish a broad framework to guide organizers and participants through the data assessment process in a datathon, regardless of their datathon structure or type of data science projects/data considered. As the reviewer points out, this is currently lacking in the literature for datathons. From a practical perspective, our framework can help focus discussions with “challenge owners” prior to datathons, e.g. on dimensions where scores are below “functional”, and helps focus the efforts of more senior research support (e.g. DSG PIs) in the refinement phase. More granular recommendations on data sufficiency or “what to do if reliability is developing and readiness is optimal“ would require an analysis on a reduced set of datathon challenges of a similar nature, which would build on this initial contribution but is beyond its scope. It is certainly a direction we hope to explore in future work.
>
> We have now made it more explicit in Section 6 how our framework can help in each phase, and added a couple of sentences on the difficulty of having more specific yet broad-purpose recommendations.
>
> 2. Incorporating Literature Recommendations
>
> We tackle the various points raised by the reviewer below.
>
> We thank the reviewer for the questions “what makes preparing data for a hackathon different from preparing data for an analysis?” and “what unique concerns does it invite?”, as they help to clarify our contribution and scope. Data preparation for datathons differs from data preparation for research projects as, over a short time, data needs to be prepared and subsequently used to answer a specific set of questions (see L19-20 in Section 1, which we have now better clarified).
>
> In research projects, there is significantly more time and room to explore and adapt the data and the questions. Identifying the few key necessary dimensions for data preparation is very valuable under datathon constraints, and that is the contribution of our paper. Such standardisation also helps when datathon organising teams change, and the host institution hopes to maintain the level of the datathons. (Update on L19-29)
>
> While we build on and relate to prior work where possible, our contribution is more specific than that of Wang and Strong, 1996 (which suggests a general framework that is datathon-agnostic with a large number of dimensions) and more general than that of Schlegel et al., 2020 (which focuses on the specific problem of defection detection in manufacturing processes). Note also that our sensitivity dimension, which is crucial in many contexts (e.g., health or social policy, commercially-sensitive settings), is covered in neither. Our goal is to suggest a small number of key dimensions that can be applied to a very broad range of datathons.
>
> Thank you for highlighting Datasheets for Datasets, which has a rather different, but complementary, purpose to our paper. Datasheets for Datasets offers helpful guidelines for dataset preparation and documentation, with questions focused on aspects like dataset content, collection mechanism, pre-processing, distribution & maintenance, etc.  We on the other hand focus specifically on data quality in the context of datathons, which overlaps with a small subset of the questions in the Datasheets for Datasets paper. We also reiterate that our goal is to suggest a handful of dimensions and tiers (rather than a specific list of questions) that can help guide data assessment in datathons. We have added a citation to Datasheets for Datasets and related work in Section 2.
>
> 3. Limitations
> Beyond the limitations already mentioned in the paper, we have added a comment on the difficulty of producing specific yet broad-purpose actionable recommendations in section 6.  We have also added the NeurIPS checklist.

---

> > ### Author Response · Authors · 2023-08-19
> >
> > 4.Minor comments:
> >
> >  - Given the possibility of adding up to one page to the paper to address reviews, we found that we could address comments and recommendations without shortening 2.1, which offers some useful context on where our work falls in the wealth of fragmented work on hackathons & datathons.
> >
> >  - We agree with the reviewer, and considered this when writing the paper, however, we also couldn’t think of a better way to do this and opted for separating (as the reviewer points out) the relationship to prior work and our proposed framework. To make Section 4 self-contained in terms of the information it contains on the dimensions, we have restated the questions associated to each dimension in the subsection title, so that the only difference in content between Section 4 and 2.2 is now relation to prior work.
> >
> >  - The purpose of section 3 is to provide information about the datathons that underpin our framework and case studies, to introduce terminology we refer back to later on (e.g., challenge owner, PIs), and perhaps most importantly, to illustrate the breadth of DSGs (in terms of topics, sectors, stakeholders, etc). The framework suggested is suitable to datathons as broad as the DSG and the more specific datathons cited in our paper (e.g., restricted to healthcare, open access data, etc).
> >
> >  - We think Figure 3 is important as it allows a reader to see “at a glance” how the case studies score within our framework (eg there is at least one example of “developing” across all but 1 dimension). It is now referenced in the first sentence of Section 5, and the figure moved to the Appendix.
> >
> > Please also see the general comment for all reviewers.

---

> > > ### Comment · Reviewer_ijbY · 2023-08-27
> > > **Response to author rebuttal**
> > >
> > > Thank you for your detailed response! I have reflected for some time on the first point (actionable insights and methodological constraints) and your response. I see how this framework could be a useful tool in guiding discussions with the challenge owners, and I think being more explicit about this in Section 6 has strengthened the paper. In light of this point, I raise my score.
> > >
> > > I would like to make my question about documentation in point 2 a bit more clear, as I don't think I was clear in my original review (my apologies for that!). The current revision of the paper contains a short paragraph in Section 2.1 with references to datasheets and model cards. Rather than just adding these citations to the paper (and as an aside, they feel slightly out of place in Section 2.1), what I was trying to encourage the authors to do was add a discussion about the relationship between data documentation and data assessment, both in terms of preparation and deciding if a dataset is ready for a datathon, and in terms of the documentation that is to be shared with the datathon participants. Anecdotally, this is something that I have found to be commonly overlooked in datathons and is often a source of confusion. Furthermore, dataset documentation is an area in which ML in particular has been found to be lacking. Datasheets are one way of providing such documentation, but at a higher level, I would like to see a more explicit discussion of where the authors see this fitting into the assessment framework. Perhaps this would go under data "readiness", where there is some mention of an optimal dataset as having a full data dictionary. I'm open to further raising my score pending author response on this point as well!

---

> > > > ### Author Response · Authors · 2023-08-28
> > > >
> > > > Many thanks to the reviewer for the thoughtful comments and for revising the score of our submission.
> > > > Adding an explicit link to data documentation (and some associated references) in Section 4.2 (Data readiness) is an excellent idea. We have now removed the citations from 2.1 and rather added references and comments on data documentation to the introduction of 4.2 and to the text for the “Optimal” tier.
> > > > This is something we might further improve upon internal deliberation in the following days on both the paper and our processes.

---

> > > > > ### Comment · Reviewer_ijbY · 2023-08-29
> > > > >
> > > > > Great, thank you for doing that; I think it strengthens the paper. I do hope you continue to deliberate on this point as you reflect upon the paper and your processes. I will raise my score an additional point.

---

### Official Review · Reviewer_ammT · 2023-07-20
**Reviews of and recommendations for datasets for datathons**

**Rating:** 7
**Confidence:** 2
**Correctness:** Yes

**Strengths:**

The authors have a unique perspective from organising as many datathons. They provide a useful framework but also use it to analyze ten datathons, which should be valuable for new organizers of datathons. The impact and relevance to the broader research community is hard to estimate: most people will not organize datathons (or maybe not even attend them), but improved quality of datathons might lead to better datasets (or understanding thereof) which have the potential to be useful in research (in addition to the inherent value the datathons provide for the challenge owner and participants).

**Additional Feedback:**


I can not find the neurips checklist.

**Clarity:**

Yes, minor errors:
 - L190 citation should have parentheses.
 - L280 should have space after (s)


**Documentation:**

Yes, the authors provide references to the website with reports on the various datathons they analyze. However, I can not find the neurips checklist.

**Ethics:**

No.

**Limitations:**

It would be good to apply the framework to a datathon outside of the Alan Turing Institute (though I see no technical limitations for doing so).

**Opportunities For Improvement:**

It would be interesting to highlight the differences to what you expect/require of a datathon dataset versus a regular dataset.
The distinction between _insufficient_ and _developing_ for the _sufficiency_ criteria (4.5) is unclear to me, especially since "hinders the overall success" of the _insufficient_ criteria is vague, and should apply to things like not being able to do meaningful statistical tests. Sections 4.1-4.3 included examples for the criteria, perhaps adding them to 4.4 and 4.5 might help understand the distinction.




**Relation To Prior Work:**

Yes

**Summary And Contributions:**

The authors share experiences of datathons they ran and provide a framework to analyse datasets for datathons in five distinct dimensions. They evaluate ten recent datathon datasets with the framework, and provide recommendations for preparing datathons. These include active engagement of challenge owners before the datathon and additional data checks before the event to possibly adjust the scope or goal of  the datathon.

---

> ### Author Response · Authors · 2023-08-16
> **Rebuttal**
>
> We thank you for the time and effort invested in evaluating our paper and for the comments which will help us improve the paper and clarify the contribution. To clarify the distinction between insufficient and developing, we have further clarified and added examples of the criteria to section 4.4. and 4.5.
>
> With respect to the limitation regarding applying the framework outside the Alan Turing Institute (ATI), we have also run datathons with ATI academic partners. For example, the data study group upcoming at the University of Exeter (https://www.exeter.ac.uk/events/details/index.php?event=13087) or past at the University of Bristol (https://www.bristol.ac.uk/golding/partner-with-us/the-alan-turing-institute/bristol-data-study-group/).
>
> Many thanks for pointing out these points; they help to improve the paper.
>
> Furthermore, following other reviewers' feedback, we have also clarified what makes a “what makes preparing data for a hackathon different from preparing data for an analysis?” and “what unique concerns does it invite?”. L19-21
>
> We have also added the NIPS checklist at the end of the paper.

---

> > ### Comment · Reviewer_ammT · 2023-08-20
> >
> > Thank you for taking the time to respond. The adjustments are satisfactory.

---

### Official Review · Reviewer_CNLi · 2023-07-21
**How to Data in Datathons**

**Rating:** 6
**Confidence:** 4
**Correctness:** NA
**Clarity:** The paper is in the top 20% of submis…

**Strengths:**

The paper's strengths are in its practicality and case studies. It addresses real-world challenges faced during datathons, offering a high level of applicability for event organizers with concrete guidelines. Moreover, the authors introduce their own framework that assesses data from several dimensions such as appropriateness, readiness, reliability, sensitivity, and sufficiency. This multidimensional approach presents a practical path for other data hackathon organizaers. Lastly, the paper provides detailed analysis of ten datathon use cases, showing how this theoretical framework can be applied in real-world scenarios.



**Additional Feedback:**

The authors have extensive history of staging datathons and thinking about the data practices. But as the objective of this paper is " to provide guidelines and recommendations that serve as a resource for organizers to navigate the data-related complexities of datathons," it is a major oversight to not include any considerations of any of the ethical complexities of working with data in such contexts.



**Documentation:**

NA

**Ethics:**

Not directly. Indirectly, the paper does not adequately address any issues regarding the ethics of data in datathon contexts.

**Limitations:**

This paper leaves out any substantial ethical issues that arise in datathons, which is a fundamental and ongoing issue for organizers. The authors just note that "they recommend all datathon projects are assessed for data protection and ethical considerations,[but] these processes are outside of the scope of this paper." As there is no details given as to why they are outside the scope, it leaves the rest of the framework with a considerable gap, and provides no guidance in this area. As the aim of this paper is to "improve the handling of data for organisations prior to datathon events", in order to be accepted it would need to have a detailed section around good data stewardship, including issues of privacy, ethics, and so on.

**Opportunities For Improvement:**

The paper  could be improved by a deeper analysis of what is at stake in the datathon model, and examples of where it has make material contributions and where it has encountered problems. While the paper provides valuable guidelines and a framework, it doesn't offer a granular quantitative evaluation of their effectiveness in any close detail. The inclusion of empirical evidence demonstrating how the proposed methods improve data handling in datathons could further strengthen the paper, also with more detailed assessments from participants.  Lastly, while several examples are provided, the process of how the challenges are mapped to the framework is not explicitly clear. This lack of clarity could pose difficulties for others trying to replicate the application of the framework.

**Relation To Prior Work:**

The paper could include more analysis of existing literature on the limitations of datathons, and best ethical practices of working with data in these contexts.

**Summary And Contributions:**

The paper addresses the challenges related to data in datathons (data science-centered hackathons), such as appropriateness, quantity, and sensitivity of data. The authors aim to establish guidelines for preparing various types of data for these events based on their extensive experience. The short timeframe and problem-specificity of datathons make data preparation challenging, further complicated by different types of datathons and data sources. To address this, the authors propose a standardized process and definitions for data selection and preparation. They introduce a framework that analyses datathon data from several dimensions including appropriateness, readiness, reliability, sensitivity, and sufficiency. Insights and experiences from organizing datathons with external organizations are shared, with 10 data challenges mapped to the proposed framework and using Data Study Groups. The paper also offers best practices for organizers to select, prepare, and categorize data for datathon challenges. The authors demonstrate the applicability of their framework with ten datathon use cases, and concludes with recommendations on improving data quality.

---

> ### Author Response · Authors · 2023-08-19
>
> We thank the reviewer for their thoughtful comments on our analysis framework and would like to commend their suggestions for improving the quality of our contribution. See also the general reply for some of the reviewers answer.
>
> We agree that a substantial part of the datathons is ethics. On the other side, ethics is not an additional data dimension in our process; it is a separate step that is comparable in size to data assessment (the focus of our paper). Also, it varies by country, organising institution and partners and needs in-depth discussion even within fields to be useful in particular, we conduct thorough and iterative ethics reviews that build on institutional guidance and guidance by Turing's Public Policy programme. We have now clarified this in the paper and plan to write follow-up research focused solely on ethical considerations in datathons.
>
>  Data privacy falls under section 4.4. L251 “Data sensitivity refers to the level of confidentiality and privacy”
>
> The reviewer also asked for an evaluation of the effectiveness of our framework. While this could require evaluation methodologies such as randomised controlled trials, it remains out of scope for this paper.  We have added a comment on the difficulty of producing specific yet broad-purpose actionable recommendations in section 6.   While still qualitative, we believe this gives the reader a better idea about the effectiveness of using the proposed five data dimensions to evaluate data for hackathons.
>
> Finally, as the reviewer recommended, we’ve added several citations on the limitations of datathons and the best ethical practices for data handling in the introduction.

---

### Official Review · Reviewer_s7kZ · 2023-07-21
**An actionable framework for improving datathon datasets.**

**Rating:** 8
**Confidence:** 3
**Correctness:** Yes
**Clarity:** Yes

**Strengths:**

The framework is straightforward and actionable.

The paper is anchored in real-world insights from the DSG. The 10 case studies are also publicly documented.

The problem of building datasets that empower time-bound, impactful problem solving in the context of hackathons is important.

**Additional Feedback:**

I really enjoyed this paper. Thank you for this work!

**Documentation:**

Not applicable

**Limitations:**

The focus of the paper is on organizing more effective and beneficial hackathons. I don’t consider an analysis of negative impacts necessary.

The recommendations section serves as helpful practical guidance for appropriate and effective use of the framework.


**Opportunities For Improvement:**

The scope of the framework is very narrow, e.g. leaving out assessment for “ethical considerations”. This narrowness helps keep the framework extremely straightforward and actionable. But it also potentially decreases the significance of the paper’s contribution.


**Relation To Prior Work:**

Yes

**Summary And Contributions:**

The paper makes a contribution to scholarship on best practices in data science hackathons (‘datathons’). The paper provides an actionable framework for evaluating and developing hackathon datasets that lower the barrier to solving real world problems in the time bound context of a datathon. The paper is informed by the practical experience the “Data Study Groups (DSGs) (“an award-winning collaborative datathon event organised by The Alan Turing Institute, the UK’s national institute for data science and artificial intelligence.”) After presenting the framework, the authors go on to apply it to multiple real-world case studies from the DSG.

---

> ### Author Response · Authors · 2023-08-16
>
> We would like to thank the reviewer for taking the time to evaluate our paper on best practices in data science hackathons. We appreciate the positive feedback and constructive criticism provided in the review.
>
> Please see general comments for paper updates and discussion.

---

### Author Response · Authors · 2023-08-19

Dear reviewers,

We would like to thank the reviewers for their comments and suggestions. All of the reviews are thoughtful and constructive. The feedback has improved the paper and the communication of our work. Thank you to all the reviewers for putting in so much time and effort. This has been a very positive review experience for our team.

In this overarching response, we highlight one common point across the reviews and one discrepancy across them. The remainder of our responses are in separate comments.

1. Consideration of ethics for datathon events.

A common discussion about the paper has been around datathon ethics.

Upon internal deliberation, we have determined that ethics constitutes not an additional dimension within our outlined process but rather a distinct step akin in scope to data assessment, which forms the central focus of our paper. Ethical considerations vary across countries, organising institutions, and collaborative partners, necessitating in-depth deliberation that even extends within specific fields to ensure practicality.

For Data Study Groups, our approach involves comprehensive and iterative ethics reviews, which draw upon institutional guidelines and the directives laid out by the Alan Turing Institute's Public Policy programme. Datathon ethics are distinct from data and project ethics, as they involve specific considerations due to the collaborative, time-limited nature, potential societal impact, and diverse range of participants and stakeholders.  We have now explicitly clarified this aspect within the paper (last paragraph of section 3, lines 193-201). Moreover, we are planning a subsequent publication that will exclusively concentrate on the nuanced ethical aspects about datathons.

For instance, a datathon centred around utilising AI for improving weapons accuracy or public surveillance might be deemed acceptable within certain institutions yet could potentially misalign with the objectives and values upheld by DSGs. It is important to note that the subject of data privacy has been addressed within the context of data sensitivity, as delineated in Section 4.4 of the paper.

2. Novelty and usefulness of a framework for assessing data for datathon events.

All of the reviewers commented that our work would be useful for anyone running datathon events. Some note that the events are only relevant for a subset of the data science and AI research and practitioner community but still agree that a framework such as ours would be useful for those teams.

All the reviewers have understood our work, and we hope our edits based on their feedback will help future readers apply the framework to their events.

Summary of updates:
- Update on ethics. (L194-202)
- Updates on recommendations to make the framework more actionable. (Section 6)
- Examples in sections 4.4 and 4.5 and category delineation improvements.
- Extended related work. (L54-58)
- Move the heatmap figure to the appendix to include more content in the main body of the paper.
- References to figures.
- Section titles update. (Section 4)
- NeurIPS checklist
- Minor grammar improvements.

---

### Decision · Program_Chairs · 2023-09-22

**Decision:**

Accept (Poster)

**Comment:**

This paper introduces a framework for preparing datasets for use in datathons. The authors distill these guidelines from years of datathon organization experience. The framework is clear, actionable, somewhat limited in scope as noted by several reviewers, though the proposed changes from the authors make progress towards addressing these limitations.

Pros:
- Introduces a clear, actionable framework for preparing/selecting datasets for datathons
- Distilled from real-world insights
- The framework would be a useful resources for any organizations or individuals striving to organize data science hackathons or similar events

Cons:
- Limited in scope: does not include recommendations re: datathon ethics, impacts of datathons etc (addressed in rebuttal)
- Limited empirical evidence on whether the proposed guidelines/methods would improve datathons (out of scope for this work)

Major issues have been addressed by the authors and any remaining comments can be addressed in a camera-ready revision.